# H3K27M induces defective chromatin spread of PRC2-mediated repressive H3K27me2/me3 and is essential for glioma tumorigenesis

Ashot S. Harutyunyan [1], Brian Krug[1], Haifen Chen[1], Simon Papillon-Cavanagh[1], Michele Zeinieh[1], Nicolas De Jay[1,2], Shriya Deshmukh[1], Carol C.L. Chen[1], Jad Belle[1], Leonie G. Mikael[3], Dylan M. Marchione[4], Rui Li[1], Hamid Nikbakht[1], Bo Hu[1], Gael Cagnone [1], Warren A. Cheung[1,5], Abdulshakour Mohammadnia[1], Denise Bechet[1], Damien Faury[1], Melissa K McConechy[1], Manav Pathania[6], Siddhant U. Jain[7], Benjamin Ellezam [8], Alexander G. Weil[9], Alexandre Montpetit[10], Paolo Salomoni[11,6], Tomi Pastinen[1,5], Chao Lu [12], Peter W. Lewis [7], Benjamin A. Garcia[4], Claudia L. Kleinman[1,2], Nada Jabado [1,3] & Jacek Majewski [1,10]

Lys-27-Met mutations in histone 3 genes (H3K27M) characterize a subgroup of deadly gliomas and decrease genome-wide H3K27 trimethylation. Here we use primary H3K27M tumor lines and isogenic CRISPR-edited controls to assess H3K27M effects in vitro and in vivo. We find that whereas H3K27me3 and H3K27me2 are normally deposited by PRC2 across broad regions, their deposition is severely reduced in H3.3K27M cells. H3K27me3 is unable to spread from large unmethylated CpG islands, while H3K27me2 can be deposited outside these PRC2 high-affinity sites but to levels corresponding to H3K27me3 deposition in wild-type cells. Our findings indicate that PRC2 recruitment and propagation on chromatin are seemingly unaffected by K27M, which mostly impairs spread of the repressive marks it catalyzes, especially H3K27me3. Genome-wide loss of H3K27me3 and me2 deposition has limited transcriptomic consequences, preferentially affecting lowly-expressed genes regulating neurogenesis. Removal of H3K27M restores H3K27me2/me3 spread, impairs cell proliferation, and completely abolishes their capacity to form tumors in mice.

---

[1] Department of Human Genetics, McGill University, Montreal, QC H3A 1B1, Canada. [2] Lady Davis Research Institute, Jewish General Hospital, Montreal, QC H3T 1E2, Canada. [3] Department of Pediatrics, McGill University, and The Research Institute of the McGill University Health Center, Montreal, QC H4A 3J1, Canada. [4] Department of Biochemistry and Biophysics, and Penn Epigenetics Institute, Perelman School of Medicine, University of Pennsylvania, Philadelphia, PA 19104, USA. [5] Center for Pediatric Genomic Medicine, Children's Mercy Kansas City, Kansas City, MO 64108, USA. [6] Samantha Dickson Brain Cancer Unit, University College London Cancer Institute, London WCE1 6DD, United Kingdom. [7] Department of Biomolecular Chemistry, School of Medicine and Public Health and Wisconsin Institute for Discovery, University of Wisconsin, Madison, WI 53715, USA. [8] Department of Pathology, Centre Hospitalier Universitaire Sainte-Justine, Université de Montréal, Montréal, QC H3T 1C5, Canada. [9] Department of Pediatric Neurosurgery, Centre Hospitalier Universitaire Sainte-Justine, Université de Montréal, Montréal, QC H3T 1C5, Canada. [10] McGill University and Genome Quebec Innovation Centre, Montreal, QC H3A 0G1, Canada. [11] Nuclear Function in CNS pathophysiology, German Center for Neurodegenerative Diseases, 53127 Bonn, Germany. [12] Department of Genetics and Development, Columbia University Irving Medical Center, New York, NY 10032, USA. These authors contributed equally: Ashot S. Harutyunyan, Brian Krug. Correspondence and requests for materials should be addressed to N.J. (email: nada.jabado@mcgill.ca) or to J.M. (email: jacek.majewski@mcgill.ca)

High-Grade Gliomas (HGG) are devastating brain tumors and a major cause of cancer-related mortality[1]. Pediatric HGG have molecular signatures distinct from those of adult HGG[2–4]. Specifically, they frequently harbor somatic mutations in histone 3 (H3) genes[5–7]. These mutations primarily impact the epigenome and show neuroanatomical and age specificity, suggesting that they occur during brain development[1,5,6,8–10]. The most frequent oncohistone, H3K27M, specifies diffuse midline gliomas, which include deadly diffuse intrinsic pontine gliomas (DIPG) and represents a newly recognized molecular HGG entity in the 2016 World Health Organization classification[11]. This somatic heterozygous mutation is present in all tumor cells at diagnosis, tumor spread, and in autopsy samples, and is recognized to be the major oncogenic driver in these HGGs[1,6,10,12,13].

The mechanism of H3K27M action remains unclear. Mutant H3K27M, which can occur in both the canonical (H3.1 or H3.2) and non-canonical (H3.3) histone variants, contributes to only a fraction of the total H3 pool (3–17%)[14]. However, it has a dominant effect as it drastically reduces overall levels of the repressive H3K27me3 mark in cells[14–16]. In vitro, H3K27M has been shown to severely affect the enzymatic activity of EZH2, a core component of the Polycomb Repressive Complex2 (PRC2), which normally catalyzes H3K27 methylation (reviewed in ref. [17]), possibly through strong binding of the enzyme to H3K27M-containing nucleosomes, which sequesters and inactivates the complex[18,19]. How the resulting in vivo loss of H3K27me3 induces tumorigenesis remains the subject of active investigations. Several contradictory hypotheses have been proposed, namely preferential recruitment and/or sequestration of PRC2 on chromatin by K27M mutant nucleosomes[14,19–21], preferential recruitment of PRC2 to its strong affinity sites[22], or exclusion of this complex by mutant nucleosomes from its normal sites in mutant cells[23]. Indeed, studies using H3K27M-DIPG lines and mouse neural progenitor cells (NPCs) manipulated to overexpress H3K27M argue that H3K27me3 loss in large genomic areas leads to increased gene expression at bivalent promoters (marked, in the normal state, by both H3K27me3 and H3K4me3)[20,24], while residual H3K27me3 deposition still occurs at several genomic loci and mediates oncogenesis[22]. Varying levels of PRC2 activity across sites have been suggested to account for these differential effects of H3K27M on distinct genomic loci, with those strongly binding PRC2 retaining the mark, and the weaker binding sites losing it in the presence of the mutation[22]. Another study suggested that the specific enrichment of H3.3K27M-carrying nucleosomes in actively transcribed genomic regions where H3.3 is preferentially deposited precludes PRC2 recruitment and subsequent H3K27me3 deposition[23]. In all, despite many enticing hypotheses, a unified view on downstream effects of H3K27M is lacking. One notable limitation of all studies to date[15,20,22,23] is the lack of an isogenic tumor-relevant context for studying the effects of the mutation. The precise cell of origin for these tumors remains unknown. Given that H3K27M is tumorigenic only when introduced in specific neurodevelopmental windows[22,23,25,26], and that H3K27me3 deposition varies with cellular context and differentiation stage[27], it follows that the absence of an appropriate isogenic background would likely represent a major confounder. In addition, no data exists on the more abundant repressive mark H3K27me2, whose production by PRC2 is similarly affected by H3K27M[14], and which may contribute to K27M tumorigenesis. Moreover, while there is evidence that H3K27M mutation is the initiating oncogenic event[12], there is a clear requirement for additional specific oncogenic partners to drive tumor formation[6,9,12,13,28–30]. Whether H3K27M is necessary for tumor progression and/or maintenance is not yet known.

To address how decreased H3K27me3/me2 levels lead to tumorigenesis and to examine the requirement of K27M for tumor maintenance, here we use multiple human primary pediatric HGG cell lines as a controlled tumor-relevant setting for the H3K27M mutation in this study. We directly manipulate H3K27M by overexpressing it in wild-type lines or using CRISPR editing to remove it from mutant cell lines. We further alter PRC2 by overexpression or pharmacological inhibition. We use epigenome and transcriptome analysis to study the molecular consequences, cell assays to profile proliferation, and a mouse orthotopic xenograft model to determine the final effect of the mutation on tumorigenicity (Supplementary Fig. 1, Supplementary Table 1). Our data show that PRC2 deposition and propagation on the chromatin are unaffected by K27M. The main effect of the mutation is in preventing the spread of H3K27me3, and to a lesser level H3K27me2, from PRC2 binding sites to larger silencing domains. This epigenetic defect is reversible upon removal of the H3K27M mutation, suggesting possible reversibility of its functional consequences. Importantly, we show H3K27M is essential for tumor maintenance as removal of this mutation in HGG is sufficient to mitigate tumor growth in vivo.

## Results

### H3K27me2 loss is less marked than H3K27me3 in H3.3K27M.

We used mass spectrometry to assess H3K27 methylation levels in primary HGG cell lines carrying H3.3K27M ($n = 3$) or wild-type ($n = 3$) for this mutation. Massive loss of H3K27me3 was observed as expected, while a significant but less drastic loss of H3K27me2, and a modest increase in H3.1/H3.2K27me1 were seen in H3.3K27M cell lines (Fig. 1a). We then used chromatin immunoprecipitation combined with sequencing (ChIP-seq) to profile H3K27me3 and H3K27me2 distribution in these cell lines. Exogenous chromatin spike-in (ChIP-Rx, see methods), used to quantitate signal intensity, confirmed overall decrease in H3K27me3/me2 methylation levels in H3K27M cells compared to wild-type HGGs (Supplementary Fig. 1–2, Supplementary Table 2). Analyzing genome-wide distribution of both marks showed striking differences between K27M and wild-type HGG lines (Fig. 1b, Supplementary Fig. 3a). While H3K27me3 was distributed in broad domains in wild-type cells, H3K27M cells showed sharp peaks of H3K27me3 coinciding with loss of the mark in the broad domains where it is normally found and resembling the pattern of embryonic stem-cells (ESC) (Supplementary Fig. 3a). Notably, H3K27me3 distribution was highly consistent across six H3K27M cell lines tested and comparable to that of the primary tumors from which the cell lines were derived (Supplementary Fig. 3b-d). In contrast, the H3K27me2 mark showed a broader distribution in K27M, resembling the distribution of H3K27me3 in wild-type HGGs, while in these cells not carrying the mutation this mark spread as expected genome-wide largely outside H3K27me3 domains (Fig. 1b, c). We further profiled PRC2 distribution by ChIP-seq of the core PRC2 component SUZ12. In H3K27M-mutant cells, H3K27me3 deposition showed near-complete co-occupancy with SUZ12, suggesting that H3K27me3 is restricted in these cells to PRC2 binding sites (Fig. 1b, c, Supplementary Fig. 4–5). Wild-type HGGs, in turn, showed additional deposition of H3K27me3 in broad domains with limited SUZ12 occupancy (Fig. 1b, c, Supplementary Fig. 4–5). Our data indicate that, in the presence of K27M, H3K27me2 production by PRC2 is decreased but to a lesser magnitude than H3K27me3. Accordingly, a level of spread on chromatin is seemingly achieved for this mark while H3K27me3 deposition appears confined to specific genomic loci.

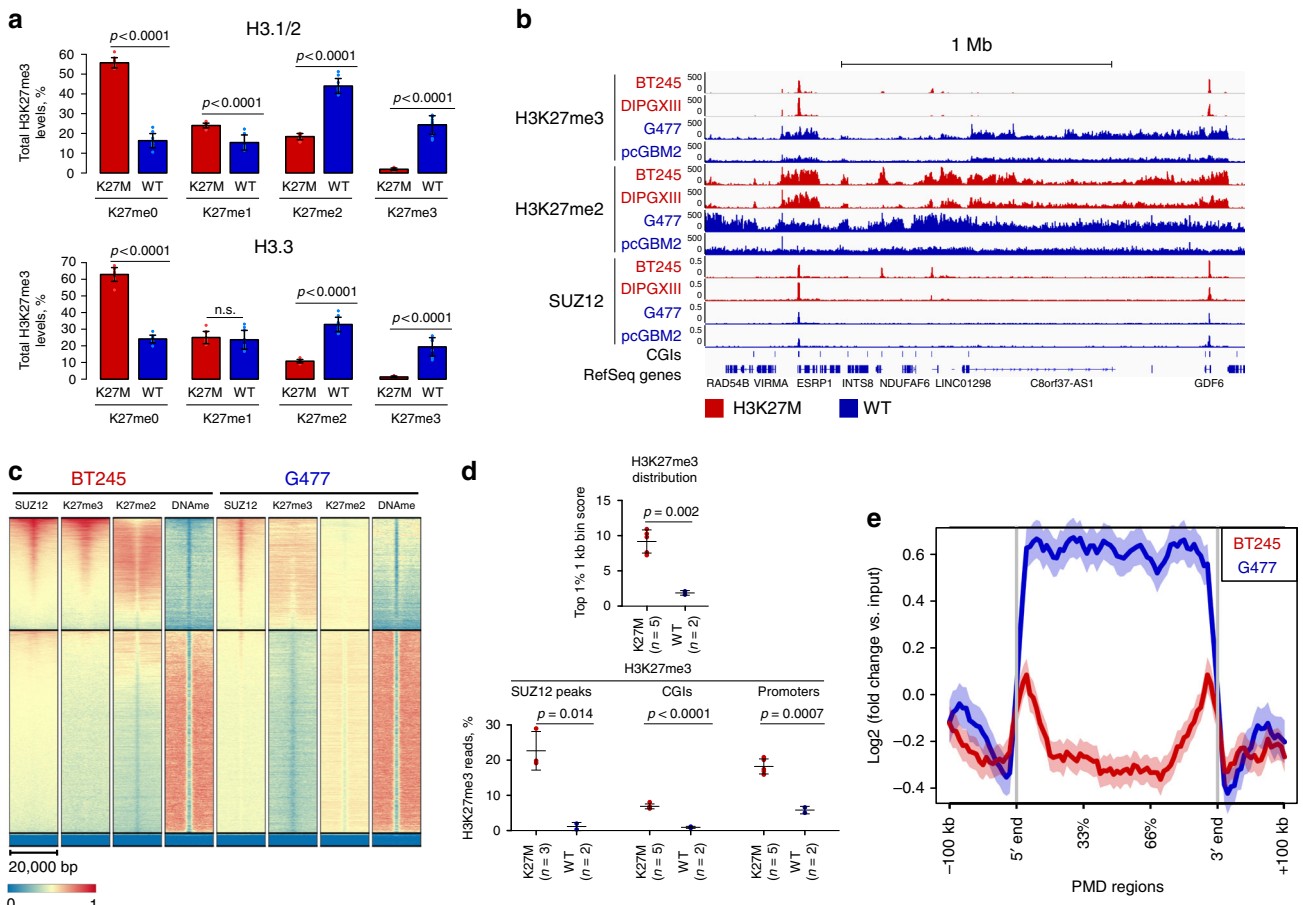

**Fig. 1** H3K27M-mutant pediatric high-grade gliomas (pHGGs) exhibit distinct distribution of H3K27me3 and H3K27me2 **a** H3K27me1/2/3 abundance quantified by mass spectrometry, primary cells (WT $n = 3$ cell lines, K27M $n = 3$ cell lines, three replicates for each cell line, mean ± standard deviation, Student's $t$-test). **b** Example of normalized H3K27me3 and H3K27me2 chromatin immunoprecipitation sequencing (ChIP-seq) tracks of H3K27M and WT pHGG lines, showing qualitative differences in distribution of these marks. For comparison, SUZ12 ChIP-seq tracks, and CpG islands (CGIs) are shown. **c** Heatmap plots of ChIP-seq signal intensity for SUZ12, H3K27me2/3, and DNA methylation (whole-genome bisulphite sequencing (WGBS)) over CGIs for BT245 (H3K27M) and G477 (WT). CGIs are separated by kmeans clustering ($k = 3$). **d** Top: Average enrichment at the top 1% of 1 kb bins for H3K27me3. Bottom: Proportion of H3K27me3 reads in SUZ12 peaks, CGIs, promoters. H3K27M cells show higher enrichment for H3K27me3 in top 1% 1 kb bins, SUZ12 peaks, CGIs and promoters compared to wild-type cells (mean ± standard deviation, Student's $t$-test). **e** H3K27me3 signal intensity over partially methylated domain (PMD) regions in BT245 (H3K27M) and G477 (WT) cells, aggregate plot. Source data are provided as a Source Data file

## H3K27me3 is retained at unmethylated CGIs.

To obtain a quantitative characteristic of H3K27me3 distribution, we divided the genome into 1 kb bins and calculated the average normalized read counts in top 1% of the most enriched bins (Fig. 1d). H3K27M cells had significantly higher enrichment in top bins, reflecting focused distribution of H3K27me3 mark in those cells in contrast to broad distribution in wild-type cells. Large unmethylated CpG islands (CGIs) are known high-affinity recruitment sites for PRC2. We subsequently quantified the proportion of H3K27me3 reads within CGIs and SUZ12 peaks and observed a significantly higher percentage of the H3K27me3 signal within those features for H3K27M cells compared to wild-type (Fig. 1d). While H3K27me3 was preserved at unmethylated CGIs (Fig. 1b, c), we noticed that broad domains of H3K27me3 in wild-type cells, generally coinciding with partially methylated DNA domains (PMD), were greatly depleted in K27M cells (Fig. 1e, Supplementary Fig. 6). Accordingly, we observed a shift in a proportion of H3K27me3 from intergenic to promoter regions between wild-type and H3K27M cells (Fig. 1d, Supplementary Fig. 7).

We next investigated the distribution of the H3K27me3 writer (PRC2) and reader (PRC1) complexes. In all cell lines, PRC2 (SUZ12) was largely localized to unmethylated CGIs (Fig. 1b, c).

However, a visibly broader pattern of SUZ12 around CGIs was observed in H3K27M cells, contrasting with its narrow deposition in wild-type lines (Fig. 1c, Supplementary Fig. 4) and consistent with an increased retention of PRC2 around its binding sites. The PRC1 complex can recognize H3K27me3 and mediate transcriptional repression but, depending on subunit composition, it can also bind and regulate active promoters[31]. We observed that the core PRC1 subunit RING1B showed increased overlap with H3K27me3/SUZ12 occupancy at CGIs in H3K27M lines compared to wild-type (Supplementary Fig. 5).

## H3K27M directly affects H3K27me3/me2 spread.

We derived H3K27M mutants by CRISPR-Cas9 in HEK293T (Supplementary Fig. 8) and overexpression of H3.3K27M in a wild-type HGG line (G477, Fig. 2a–c, Supplementary Fig. 9). In both lines, we observed significant loss of H3K27me3 (Fig. 2a–c, Supplementary Fig. 8–9) and conversion of broad domains to narrow peaks, resembling patterns seen in primary H3K27M-HGG lines (Fig. 2b, c, Supplementary Fig. 8–9). Accordingly, the proportion of H3K27me3 increased in CGIs and promoters (Supplementary Fig. 8–9). Impaired spreading of methylated H3K27 upon K27M expression is consistent both in wild-type HGG cells and

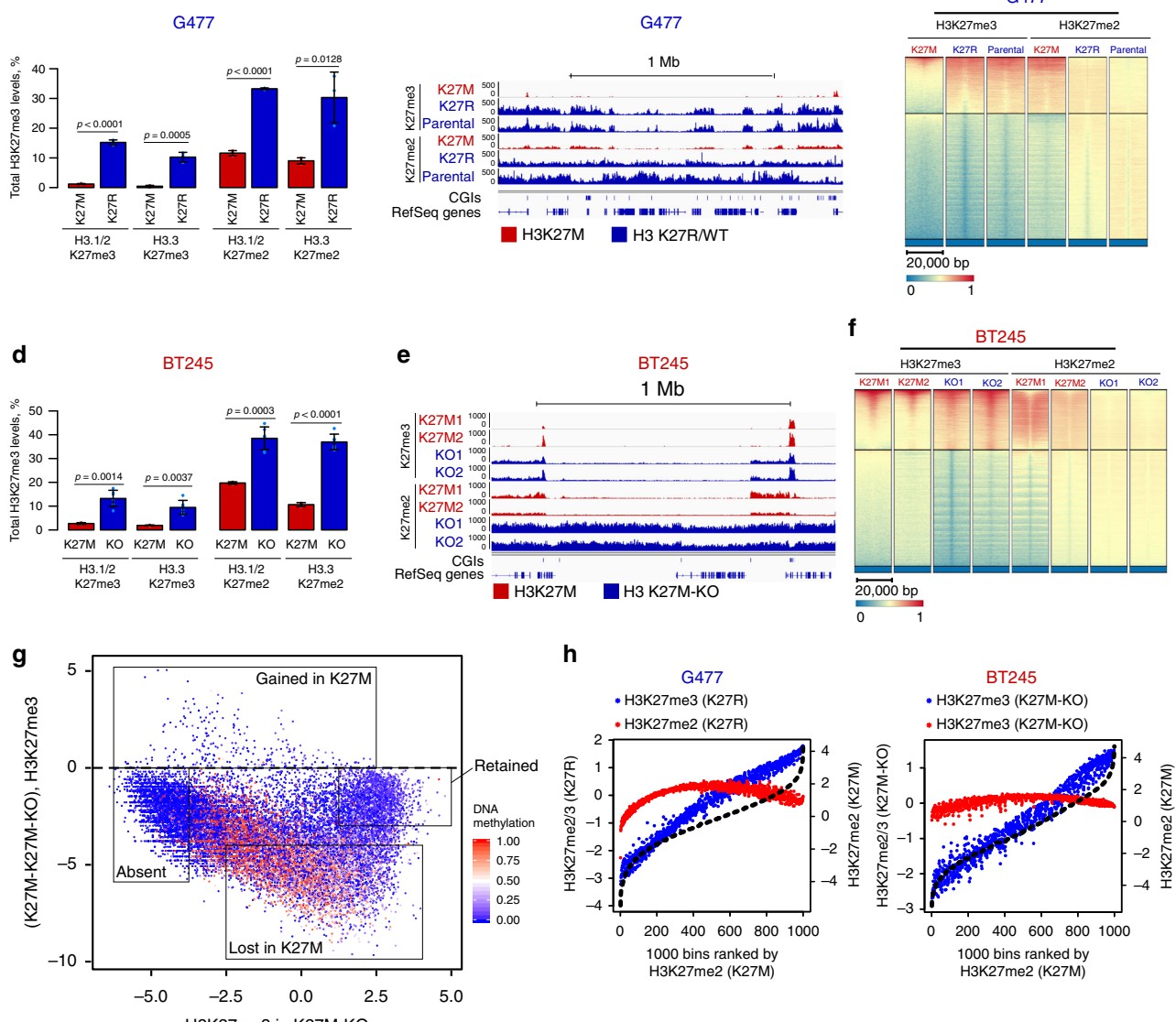

**Fig. 2** H3K27M reversibly induces global redistribution of H3K27me2 and H3K27me3 (**a–c**). G477 wild-type cell line with H3.3-K27M overexpression. **a** H3K27me2/3 abundance by histone mass spectrometry (n = 3 in each group, mean ± standard deviation, Student's t-test). **b** Example ChIP-seq tracks of H3K27me2/3 distribution, normalized by drosophila spike-in (ChIP-Rx). **c** Heatmap plots of ChIP-seq signal intensity for H3K27me2 and H3K27me3 over CGIs for parental G477 (wild-type) and G477 overexpressing K27R and K27M. CGIs are separated by kmeans clustering (k = 3). **d–f**. BT245 with H3K27M knockout (KO) by CRISPR/Cas9. **d** H3K27me3 abundance by histone mass spectrometry (K27M n = 3, KO n = 6, mean ± standard deviation, Student's t-test). Whiskers represent standard deviation. **e** Example ChIP-seq tracks of H3K27me2/3 distribution, ChIP-Rx normalized. **f** Heatmap plots of ChIP-seq signal intensity for H3K27me2 and H3K27me3 over CGIs for parental BT245 (H3K27M) and K27M knockout (K27M-KO) by CRISPR. CGIs are separated by kmeans clustering (k = 3). **g** H3K27me3 signal change at CGIs of BT245, K27M-KO vs. K27M, color coded for DNA methylation. y-axis shows the difference in normalized H3K27me3 levels at CGIs in K27M vs. K27M-KO (log2), while x-axis shows normalized H3K27me3 levels in non-K27M state (K27M-KO, log2). Four categories of CGIs based on H3K27me3 levels and difference are depicted by squares. **h** Strong correlation of H3K27me2 in H3K27M with H3K27me3 in respective isogenic non-K27M state (K27R in G477, K27M-KO in BT245). 1000 aggregate bins are ranked on x-axis based on H3K27me2 in H3K27M state (black dotted line). H3K27me3 levels in each bin for non-K27M sample are shown in blue, H3K27me2 levels in non-K27M—in red (normalized, log2). Source data are provided as a Source Data file

HEK293T, an unrelated differentiated cell type, demonstrating this effect is independent of cell context and is directly linked to the mutation. Interestingly, when assessing H3K27me2 deposition in H3.3K27M-expressing G477, this mark showed significantly less broad deposition, while its genome-wide distribution was strikingly similar to that of H3K27me3 in the parental G477 line (Fig. 2b, c). This further supports our observation that the effect of K27M appears more pronounced on H3K27me3 deposition than on H3K27me2.

To study the mutation in a controlled, tumor-relevant context, we removed the K27M mutant allele from two HGG lines, BT245 and DIPGXIII (SU-DIPGXIII), using the CRISPR/Cas9 system (Supplementary Fig. 10–11). Edited clones (H3K27M-KO) showed an average five-fold increase in H3K27me3 and a 2–3-fold increase in H3K27me2 levels (Fig. 2d, Supplementary Fig. 10–11). This was accompanied by spread of H3K27me3 and H3K27me2 signals into broad domains, resembling those of wild-type HGG cell lines (Fig. 2e, f, Supplementary Fig. 10–11),

and subsequent decrease in the proportion of H3K27me3 reads in promoters and CGIs (Supplementary Fig. 10–11). A number of unmethylated CGIs (Fig. 2g, Supplementary Fig. 10, 12–13) had low to absent H3K27me3 levels both in the original mutant cell lines and in isogenic H3K27M-KO, which generally corresponded to active promoters marked by H3K27ac (absent). Also, a number of unmethylated CGIs with no H3K27me3 in H3K27M cells (lost) showed deposition of the mark in H3K27M-KO. Notably, the degree of loss in H3K27M cells was positively correlated with the levels of the mark on that site in H3K27M-KO edited cells. Importantly, the large unmethylated CGIs enriched for SUZ12, where most of H3K27me3 is deposited in H3K27M cells, retained the mark (retained), which spread from these sites in H3K27M-KO cells (Fig. 2g, Supplementary Fig. 10, 12–13). The relatively broad enrichment of SUZ12 in these regions in H3K27M became more focused in H3K27M-KO cells, similar to the SUZ12 pattern observed in wild-type HGG (Supplementary Fig. 12, 14). Very few CGIs acquired higher H3K27me3 levels in the presence of H3K27M (gained); these gained regions were distinct in each edited line (Supplementary Table 3), and thus likely represent secondary downstream effects of unclear significance. Interestingly, the *CDKN2A* (p16) locus, a candidate driver gene in K27M gliomas[22], retained high-levels of H3K27me3, regardless of the presence of the H3K27M mutation (Supplementary Fig. 15). Finally, H3K27me3 retention has been proposed to correspond to regions of H3K27M deposition[19]. However, levels of H3K27me3 and H3K27M did not positively correlate in our cell lines or in a mouse model[26] of H3K27M tumorigenesis (Supplementary Fig. 16). Thus, H3K27M redistributes H3K27me3 to follow PRC2 recruitment, but not H3.3/K27M deposition. In addition, the H3K27me2 mark spreads outside CGIs in H3.3K27M mutant cells. Notably, when comparing H3K27M lines to their isogenic CRISPR-edited lines, we observe that it is confined in mutant cells to regions of H3K27me3 spread in H3.3K27M-KO lines (Fig. 2e, f, Supplementary Fig. 11) similar to what we observed in H3.3K27M G477 cells (Fig. 2b, c). Indeed, a strong correlation between H3K27me2 in K27M and H3K27me3 in wild-type isogenic cells is seen (Fig. 2h). Overall, these findings, together with our data on H3K27M-KO or H3.3K27M-expressing wild-type HGG, indicate that H3K27M decreases H3K27me3 and to a lower extent K27me2 levels and deposition. Importantly, while H3K27me3 is confined to specific and narrow regions corresponding to large unmethylated CGIs marked with PRC2, H3K27me2 is able to spread outside these domains in the presence of the mutation. Unlike H3K27me3, di-methylation of H3K27 is enzymatically easier for the PRC2 complex and can be produced by transient interactions with the chromatin[32,33], possibly accounting for the differential effects we observe for both marks in our model systems.

**H3K27M does not sequester PRC2 on chromatin.** Pediatric HGG have no known mutations in PRC2 components[6,9,34], suggesting the effect of H3K27M may be distinct from complete loss of PRC2 function. Moreover, small molecule PRC2 inhibitors impair the growth of H3K27M lines[22]. We replicated this in our model system where we treated H3K27M-HGG and isogenic H3K27M-KO cell lines using two EZH2 inhibitors and found significantly more growth inhibition in the presence of K27M (Supplementary Fig. 17). Pharmacological EZH2 inhibition drastically decreases H3K27me3 and H3K27me2 marks and uniformly affects their deposition and spread in both lines (Fig. 3a, b, Supplementary Fig. 17). In contrast, the effect of H3K27M is more pronounced on H3K27me3 deposition, which is then restricted to CGIs, therefore suggesting that the mutation does not fully phenocopy PRC2 inhibition. Next, we tested

whether we could rescue H3K27M-induced effects by over-expressing either: (1) wild-type EZH2 or (2) EZH2-Y641N, a preferential di- to tri-methylase mutant[35] shown to be less sensitive to inhibition by H3K27M[14]. Overexpression of wild-type EZH2 had minor effects on H3K27me3 and H3K27me2 (~1.5-fold gain) suggesting that increased EZH2 levels are unable to overcome K27M-induced confinement of both marks (Supplementary Fig. 18). Strikingly, EZH2-Y641N induced a large gain of H3K27me3 (~7-fold). Notably, this was achieved by the spread of H3K27me3 from sites retaining the mark in H3K27M cells on corresponding domains showing H3K27me2 spread in H3.3K27M mutant cells (Fig. 3c–e, Supplementary Fig. 18). Accordingly, a drastic depletion of H3K27me2 in H3.3K27M mutant cells overexpressing EZH2-Y641N was observed in these regions (Fig. 3c, Supplementary Fig. 18). As no new PRC2 nucleation sites were generated, our overall data further support that H3K27M induces defective spread of H3K27me3/me2, which is more pronounced for H3K27me3. Importantly, H3.3K27M does not appear to retain the PRC2 complex at CGIs, as we observe spread of H3K27me2 despite the presence of the mutation, while spread of the H3K27me3 mark is possible in this setting when overexpressing EZH2-Y641N.

**H3K27M induces limited transcriptomic changes.** To assess specific effects of H3K27M mutation on gene expression, we compared the transcriptomes of isogenic lines in the presence or absence of H3K27M (G477, HEK293T, BT245, DIPGXIII). We correlated changes in gene expression with promoter H3K27me3 levels. Although H3K27me3 is lost in wide genomic areas, the resulting effect on transcription in H3K27M-mutant cells is relatively modest. Indeed, most genes had no H3K27me3 (Absent) or no change in H3K27me3 levels (Retained) on their promoter regardless of H3K27M (Fig. 4a). As expected, we observed limited changes in their expression. Despite the loss of H3K27me3 on a large number of promoters in the context of H3K27M (Lost), only a modest fraction of associated genes showed differential expression, mainly upregulation. Moreover, very few genes had a gain of promoter H3K27me3 (Gained) in H3K27M-expressing cells, only a small fraction of which were downregulated (Fig. 4b). In all our model systems—irrespective of whether H3K27M was introduced or removed—the presence of H3K27M resulted in overall transcriptional derepression characterized by an excess of upregulated genes (Fig. 4b, c). Interestingly, the number of upregulated genes was highest within the low-expression range (Fig. 4c), suggesting that one of the effects of H3K27M on the transcriptome is to disable proper H3K27me3 repression and permit low levels of aberrant transcription. To identify potential driver targets/pathways in H3K27M mutagenesis, we investigated the genes that were significantly affected by knockout of H3K27M in both BT245 and DIPGXIII cells (Supplementary Table 4). In the presence of H3K27M, we found more upregulated (102) than downregulated (12) genes, reflecting the overall trend of gene derepression. Pathway analysis on the downregulated set did not identify any Gene Ontology (GO) enrichment, but the upregulated set showed overrepresentation of genes involved in neural differentiation and developmental pathways, consistent with previous observations in other systems[22,25,26] (Fig. 4d). Notably, across all three HGG lines, the presence of H3K27M-induced upregulation of *ID1-ID4* genes (Supplementary Fig. 19). These ID (inhibitors of DNA binding/ differentiation) genes are broadly implicated in developmental processes, inhibition of differentiation, maintenance of self-renewal, and multipotency in stem cells. ID genes have been shown to be upregulated in response to growth factor receptor stimulation, including ACVR1, a receptor activated in 20% of

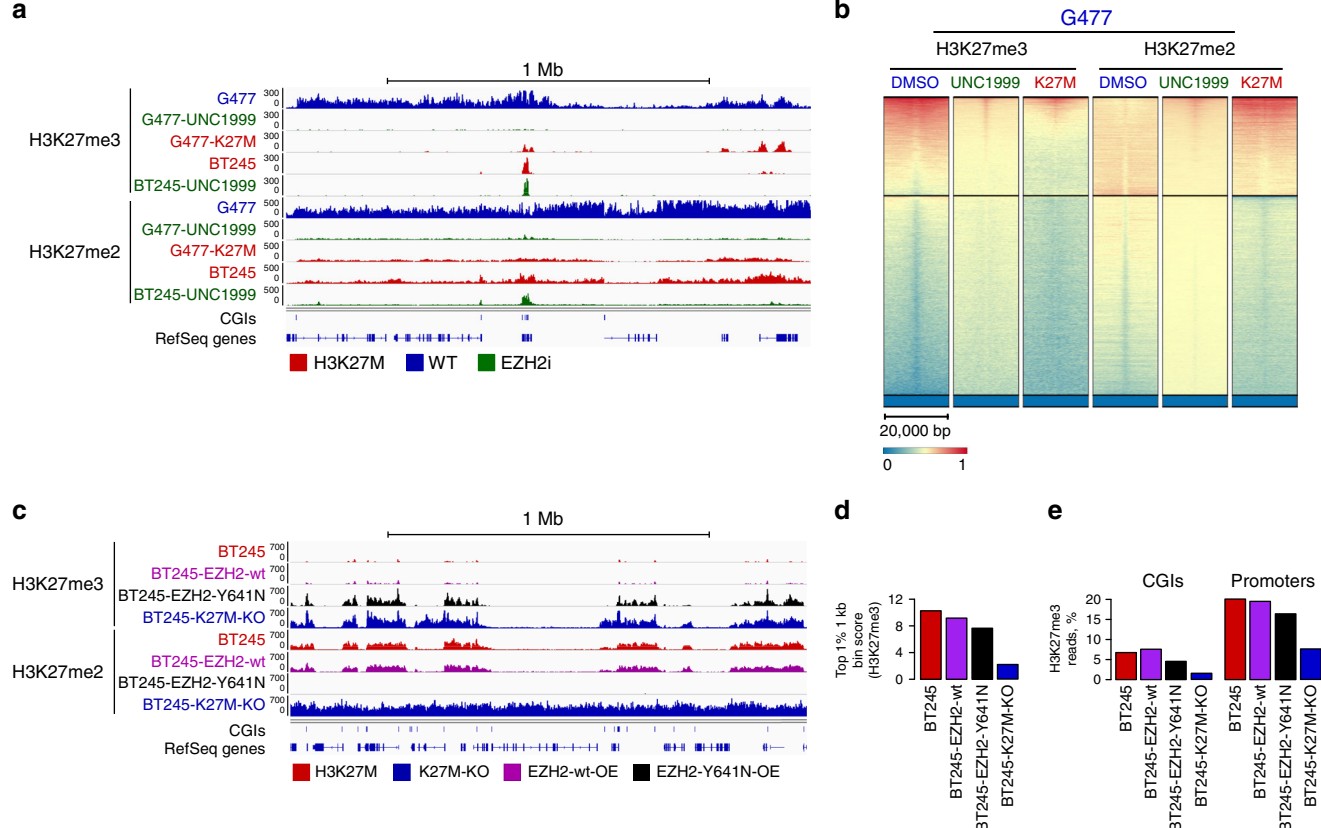

**Fig. 3** EZH2 inhibition does not completely mimic H3K27M effects and EZH2-Y641N partly overcomes H3K27M-induced effects (**a**, **b**). Changes in H3K27me2/3 levels and distribution upon EZH2 inhibition. **a** Example ChIP-seq tracks of H3K27me2/3 distribution in G477 (wild-type) and BT245 (H3K27M) cell lines treated and not treated with UNC1999 (EZH1/2 inhibitor), ChIP-Rx normalized. For comparison, G477 cell line with H3.3-K27M overexpression is also included. **b** Heatmap plots of ChIP-seq signal intensity for H3K27me2 and H3K27me3 over CGIs for G477 cell line (wild-type) treated and not treated with UNC1999, as well as with H3.3-K27M overexpression. CGIs are separated by kmeans clustering ($k = 3$). **c–e** Changes in H3K27me2/3 levels and distribution upon overexpression of EZH2-wt and EZH2-Y641N. **c** Example ChIP-seq tracks (ChIP-Rx normalized) of H3K27me2/3 distribution in BT245 cell line (H3K27M) overexpressing wild-type or Y641N mutant forms of EZH2. Parental BT245 and CRISPR-edited K27M-KO clone are included for comparison. **d** Average enrichment at the top 1% of 1 kb bins for H3K27me3. **e** Proportion of H3K27me3 reads in CGIs and promoters upon EZH2 overexpression. Source data are provided as a Source Data file

DIPGs[9,28–30]. Thus, H3K27M-HGG can promote an undifferentiated cellular state through aberrant expression of *ID* genes due to both K27M or ACVR1 mutations.

**H3K27M is required for tumorigenesis.** We examined the consequences of the introduction and removal of the H3K27M mutation on cell proliferation in vitro, as well as on tumor growth in orthotopic xenograft models (PDOX). In vitro, across the three tumor cell lines, we observed a consistent trend of H3K27M conferring a proliferative advantage (Supplementary Fig. 20a). In BT245, where H3K27M removal has the strongest effect, we were able to rescue the decrease in growth rate when we re-expressed the mutation (Supplementary Fig. 20b). To investigate the requirement of H3K27M for tumor growth in vivo, we injected independently edited BT245 and DIPGXIII (H3K27M-KO) clonal lines in the brains of NSG mice. As positive controls, for both BT245 and DIPGXIII, we performed parallel injections with two H3K27M-mutant lines (one parental and one unedited). We have previously successfully propagated BT245 and DIPGXIII as PDOX in numerous experiments and, as expected, mice injected with the positive control H3K27M lines progressively developed tumors resembling HGG. However, none of the mice injected with H3K27M-KO cells developed tumors during follow-up of over 1 year for BT245 and 6 month for DIPGXIII (Fig. 5a,

Supplementary Fig. 20c), despite the presence of other highly oncogenic mutations in the injected cells (*TP53* in both parental lines, *TERT* promoter mutation and C-MYC amplification present in the BT245 parental line, N-MYC amplification in DIPGXIII line). Our data strongly indicate that the H3K27M mutation is not only necessary for initial stages of tumor establishment but also continuously needed to preserve the proliferative and tumorigenic potential of these tumors.

**Discussion**
We provide evidence that H3K27M impairs the production and the spread of the repressive H3K27me3 mark from PRC2 high-affinity sites (Fig. 5d). We observe non-random decrease of H3K27me3 in the presence of the mutation, with residual deposition confined to large unmethylated CGIs enriched for PRC2. Notably, we show that the production and the distribution of H3K27me2, a repressive mark that is abundantly deposited by PRC2 across the silent euchromatin and the basis from which this complex synthesizes H3K27me3[32,33], are also affected by H3.3K27M, albeit at a lower magnitude than H3K27me3. Importantly, in H3.3K27M mutant cells, H3K27me2 spreads outside unmethylated CGIs, mirroring H3K27me3 distribution in wild-type cells, indicative that the PRC2 complex is not retained at these CGIs. Introducing H3K27M in HGG cell lines wild-type

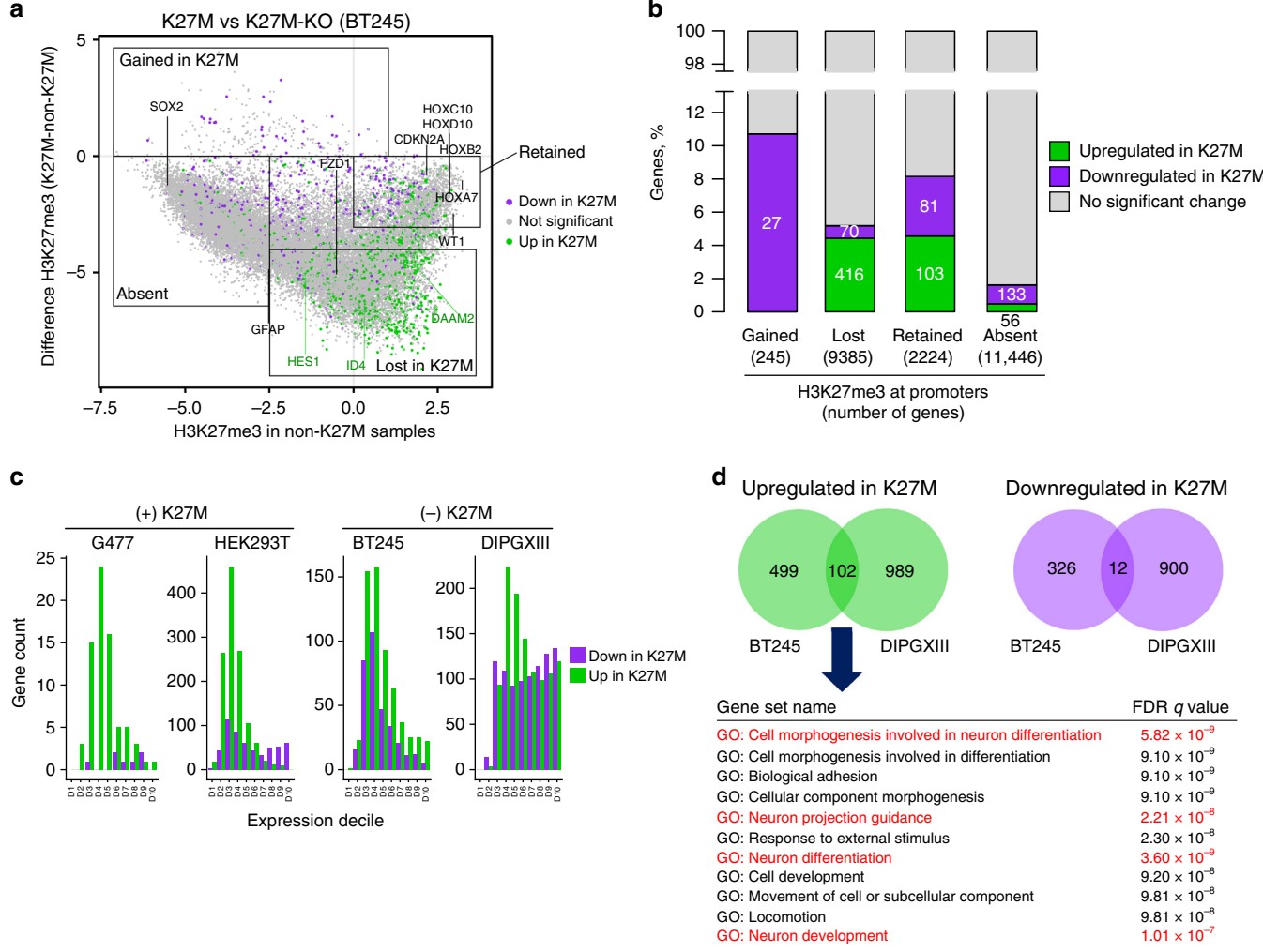

**Fig. 4** Transcriptome and H3K27me3 loci implicate H3K27M in neural differentiation **a** H3K27me3 level difference plot in BT245 dataset, color coded by gene expression changes (green for upregulated in K27M state, purple—downregulated, grey—no significant change in expression). *y*-axis shows the difference in normalized H3K27me3 levels at promoters in K27M vs. K27M-KO (log2), while *x*-axis shows normalized H3K27me3 levels in non-K27M state (K27M-KO, log2). Four categories of promoters based on H3K27me3 levels and difference are depicted by squares. **b** Number and proportion (*x*-axis) of significantly up or downregulated genes in each selected region from plot 4a (gained in K27M, lost in K27M, retained, absent). Numbers inside the columns show the number of genes up or downregulated in each category, while the number in brackets below column labels shows the total number of genes in that category. **c** Gene expression changes by deciles in different experimental datasets. The genes were assigned to deciles according to their expression in the original cell line, before manipulation. Most of differentially expressed genes (and upregulated in K27M) are among lower expressed deciles in all four datasets. **d** Overlap of differentially expressed genes in BT245 and DIPGXIII datasets. Gene set enrichment analysis of differentially expressed genes (genes upregulated in K27M state in both cell lines)

for this mutation and in HEK293T cells reproduces decreased production and confinement of both marks (H3K27me3 to CGIs, H3K27me2 to spread of H3K27me3 in wild-type cells). We further show that defective H3K27me3/me2 spread induced by K27M is reversible. When the mutation is removed, uninhibited PRC2 restores deposition of H3K27me3 from these large CGIs to sites that had lost it, mainly intergenic sites and PMDs with poor PRC2 recruitment, while H3K27me2 distribution becomes comparable to that seen in wild-type HGGs. Additionally, Y641N-EZH2, an EZH2 mutant less sensitive to H3K27M inhibition which catalyzes me2/me3[14], restores deposition of H3K27me3 by extending it from existing PRC2 recruitment sites, without creating new sites, similar to what we observe in H3K27M-KO isogenic cells. This further argues against retention of the PRC2 complex at given genomic loci and supports a K27M-induced defective spread of both H3K27me2 and H3K27me3 repressive marks.

Several recent reports on PRC2 function support our findings that decreased catalysis of K27me3/me2 due to K27M results in a defective spread on chromatin of these repressive marks from PRC2 high-affinity sites. In one study, newly deposited H3K27me3 and me2 progress from PRC2 high-affinity sites recruiting the complex following total erasure[36]. This observation is in keeping with our findings that most of the retained H3K27me3 peaks in H3K27M cells correspond to unmethylated CGIs, while H3K27me2 spreads from these sites, even though at lower magnitude compared to wild-type cells. This indicates that the PRC2 complex is not retained by the mutation and can still spread on chromatin. The differential effects observed for K27me2 and K27me3 are probably due to differential catalytic constraints for PRC2 when depositing these marks. Indeed, production of H3K27me2 is enzymatically easier and seems to require transient interactions on the chromatin for the complex as compared to H3K27me3, which is produced from H3K27me2

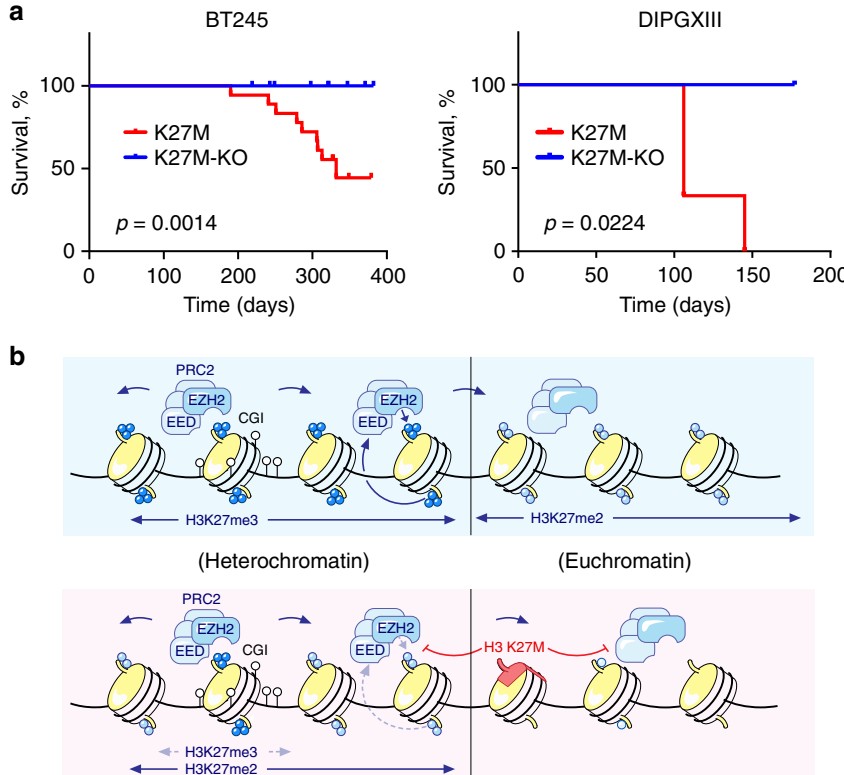

**Fig. 5** H3K27M confers tumorigenicity in vivo. **a** Survival of mouse orthotopic xenograft cohorts injected with BT245 (K27M; n = 18 mice, K27M-KO; n = 19 mice, log-rank test) and DIPGXIII lines (K27M; n = 3 mice, K27M-KO; n = 3 mice, log-rank test). **b** A model of H3K27M reversibly inhibiting the spread of H3K27me2 and H3K27me3 deposition by PRC2 from initial recruitment sites. Note that while in H3K27M H3K27me3 mark is restricted to unmethylated CGIs, H3K27me2 is found in domains where there is H3K27me3 in non-K27M condition. Source data are provided as a Source Data file

and requires a more stable association with PRC2[19,37]. This may also account for the broader distribution of SUZ12 in H3K27M-mutant cells and the near-complete overlap with H3K27me3 we observe at unmethylated CGIs. Lack of distribution of H3K27me3 from PRC2 recruitment high-affinity sites, may also be partly due to poor allosteric activation of EZH2 following decreased production of H3K27me3 in the presence of K27M, which would further preclude the spread of the repressive mark. Indeed, EED is an essential subunit of PRC2 that recognizes initial H3K27me3 and allosterically activates the complex, promoting further deposition of the mark[38]. Also, an EED cage mutant that prevents H3K27me3 recognition by PRC2 was shown to induce similar confinement of H3K27me3[39], indicating a potential effect of K27M in impairing allosteric activation of PRC2. Further experiments are needed to support decreased allosteric activation of EED as a consequence of K27M mutagenesis. In addition, the lack of H3K27me2 deposition in H3K27M cells outside typical H3K27me3 domains could be due to the presence of other histone marks such as H3K36me2 in those regions. This defect in spread of the marks, but not recruitment or spread of PRC2, is further supported by the dynamic deposition of H3K27me3 at novel loci in response to external stimuli that we (Supplementary Fig. 15 and[26]) and others[22] observe, despite the presence of the mutation. These suggest that in H3K27M cells, PRC2 could be recruited to locally as needed to deposit H3K27me3 as the effect required does not involve spread of the mark.

Experimental results obtained with the pharmacological inhibition of EZH2 indicates that the mutation does not fully reproduce pharmacological EZH2 inhibition in K27M wild-type HGG lines. In addition, specific retention of H3K27me3 in H3K27M lines suggests regional functional requirement for

H3K27me3, and may explain their sensitivity to these EZH2 inhibitors that we and others[22] observe, as these cells may not be able to tolerate further loss of the mark. Interestingly, while K27M induces widespread loss of H3K27me3, only a modest proportion of genes that lose the mark at their promoters are dysregulated. One explanation of such modest transcriptional dysregulation could be the observation that in H3K27M cells, H3K27me2 occupies the regions that lost H3K27me3 and it might maintain certain level of gene silencing. We propose that the changes observed in the presence of H3K27M consist predominantly of low-level transcriptional derepression that results from incomplete silencing. Dysregulated genes showed over-representation of functions involved in stemness and neurogenesis, in keeping with recent results suggesting that K-M mutations on H3K27 or K36 induce transcriptional deregulation affecting cell differentiation[40]. In normally developing cells, lineage commitment and further cell differentiation is mediated by a multitude of epigenetic changes, including an interplay of acquisition or loss of H3K27me3 and me2 at multiple sites, with increased H3K27me3 deposition needed upon cellular differentiation[24,27,41]. While normal cells can dynamically regulate this mark to differentiate, H3K27M-HGG have severely impaired H3K27me3 production and show its confinement to specific CGIs, not allowing the mark to fully silence neighboring regions and thus possibly stalling further differentiation. The CGIs retaining H3K27me3, from which the mark would have spread in the absence of H3K27M, may thus represent original binding sites in committed NPCs, as suggested by the blockade in neurogenesis and brain developmental pathways we observe in K27M HGG and in keeping with recent data indicating an oligodendroglial progenitor cell as a potential cell of origin for these

HGG[25,26,37,42]. Notably, our results mirror findings in ES cells, which show similar restriction of H3K27me3 to CGIs, and where loss of PRC2 does not promote major transcriptional changes but stalls further differentiation and lineage specification.

There is an ongoing debate regarding the requirement of H3K27M in maintaining tumorigenesis. Our data show that the removal of K27M is sufficient to abolish tumor formation in an in vivo orthotopic mouse model. Indeed, despite the presence of other oncogenic lesions, including *TERT* promoter mutation, *TP53* alterations, and *C-MYC* or *N-MYC* amplification, no tumors formed in the brain of mice injected with HGG cells rendered wild-type for the H3K27M mutation. The mice were followed for over 6 months to one year to exclude the possibility of delayed tumor onset. The K27M mutation is thus not simply a remnant of initial stages of tumorigenesis, it seems to be continuously required for both tumor formation and maintenance.

In summary, we propose a model where the K27M mutation affects H3K27me3 and H3K27me2 production and the relative spread of both these repressive marks from initial PRC2 binding sites in the cell of origin. Neither the recruitment of PRC2 to its nucleation sites nor the deposition of the H3K27me3 mark in the proximity of those sites is affected by the mutation. However, as the marks cannot spread to establish the proper silencing landscape, further lineage specification, a major role of PRC2, is not possible, and the cell is stalled in an early epigenetic and progenitor state, indefinitely multiplying without being able to further differentiate. Over time, this will allow acquisition of other genetic alterations ultimately leading to tumor formation. We propose that H3K27M has a vital role for maintaining this specific epigenomic landscape that prevents further cell differentiation. This in turn allows for tumor formation and is necessary for tumor maintenance, making H3K27M an important therapeutic target in HGG. Compounds targeting this mutation, when available, have the potential to greatly improve survival in this deadly cancer.

## Methods

**Patient samples and clinical information.** This study was approved by the Institutional Review Board of the respective institutions from which the samples were collected. We thank Keith Ligon and Michelle Monje for generously sharing primary tumor cell lines established from patients with high-grade glioma.

**Cell culture.** Tumor-derived cell lines were maintained in Neucult NS-A proliferation media (StemCell Technologies) supplemented with bFGF (10 ng/mL) (StemCell Technologies), rhEGF (20 ng/mL) (StemCell Technologies), and heparin (0.0002%) (StemCell Technologies) on plates coated in poly-L-ornithine (0.01%) (Sigma) and laminin (0.01 mg/mL) (Sigma). HEK293T cells (ATCC) were cultured in DMEM containing 4.5 g/L glucose, L-glutamine, phenol red, and 10% FBS (Wisent). All lines tested negative for mycoplasma contamination, checked monthly using the MycoAlert Mycoplasma Detection Kit (Lonza). Tumor-derived cell lines (Supplementary Table 1) were confirmed to match original samples by STR fingerprinting.

**CRISPR/Cas9 genome editing.** pSpCas9(BB)-2A-GFP (PX458) was a gift from Feng Zhang (Addgene plasmid # 48138). pSpCas9n(BB)-2A-GFP (PX461) was a gift from Feng Zhang (Addgene plasmid # 48140).

CRISPR-Cas9 editing was carried out as described in ref. [43]. Constructs were transfected with lipofectamine 2000 (Thermo Fischer Scientific) according to the manufacturer's protocol. Flow cytometry sorted single GFP + cells in 96-well plates, 72-h post-transfection. A summary of the FACS gating strategy is provided in the Source Data File. Clones were expanded and the target locus sequenced by Sanger sequencing. Selected clones were screened by Illumina MiSeq system for the target exon to confirm complete mutation of the K27M allele. Mass spectrometry confirmed the absence of K27M mutant peptide in these clones.

In HEK293T cells, clones heterozygous for HIST1H3B-K27M were derived through use of PX461 with the guide sequence and repair template in Supplementary Table 5. In primary HGG lines heterozygous for H3F3A-K27M, clones were derived with the mutant allele edited using PX458 and the guide sequence in Supplementary Table 5.

**Lentiviral transduction.** Lentiviruses were gifts from Dr. Peter Lewis. EZH2-WT and EZH2-Y641N constructs were applied to cells for 24 h, and puromycin (Wisent) selection (2 ug/mL) was maintained for the duration of growth. Lentiviruses expressing H3.3-K27R and H3.3-K27M were applied for 24 h and G418 (Wisent) selection (500 ng/mL) was maintained for the duration of growth.

**Western blotting.** Cells were lysed using RIPA buffer with added protease inhibitors (Roche). Whole lysate protein concentration was determined by the Bradford assay reagent (Bio-Rad). Ten micrograms of protein was separated on NuPAGE 3–8% Tris Acetate Protein gels (ThermoFischer Scientific) and wet-transferred to a nitrocellulose membrane (Bio-Rad). Membrane blocking was performed with 5% skim milk in tris buffered saline (50 mM Tris, 150 mM NaCl, 0.1% Tween 20, pH 7.4) (TBST) for 1 h. Membranes were incubated overnight with primary antibody solutions in 1% skim milk in TBST: anti-EZH2 ((1:1000, CST 5246), anti-H3K27M (1:200, Millipore ABE419), anti-H3K27me3 (1:1000, Millipore ABE44), anti-total H3 (1:2000, Abcam 1791), and anti-beta-actin (1:1000, CST 4970), also listed in Supplementary Table 6. Membranes were washed three times in TBST, and the ECL anti-rabbit IgG Horseradish Perixidase linked whole antibody (GE Healthcare) was applied for 1 h, at 1:1000 dilution in 1% skim milk in TBST. Membranes were washed three times and the signal was resolved with Amersham ECL Prime Western Blotting Detection Reagent (GE Healthcare) and imaged on a ChemiDoc MP Imaging System (Bio-Rad). Uncropped versions of all blots are provided in Source Data file.

**Histone modification identification and quantification with nLC-MS.** The complete workflow for histone extraction, LC/MS, and data analysis was recently described in detail[44]. Briefly, cell pellets (~$1 \times 10^6$ cells) were lysed on ice in nuclear isolation buffer supplemented with 0.3% NP-40 alternative. Isolated nuclei were incubated with 0.4 N H2SO4 for 3 h at 4 °C with agitation. Hundred percent trichloroacetic acid (w/v) was added to the acid extract to a final concentration of 20% and samples were incubated on ice overnight to precipitate histones. The resulting histone pellets were rinsed with ice cold acetone + 0.1% HCl and then with ice cold acetone before resuspension in water and protein estimation by Bradford assay. Approximately 20 μg of histone extract was then resuspended in 100 mM ammonium bicarbonate and derivatized with propionic anhydride. One microgram of trypsin was added and samples were incubated overnight at 37 ºC. After tryptic digestion, a cocktail of isotopically-labeled synthetic histone peptides was spiked in at a final concentration of 250 fmol/μg and propionic anhydride derivatization was performed for second time. The resulting histone peptides were desalted using C18 Stage Tips, dried using a centrifugal evaporator, and reconstituted using 0.1% formic acid in preparation for nanoLC-MS analysis.

nanoLC was performed using a Thermo ScientificTM Easy nLCTM 1000 equipped with a 75 μm × 20 cm in-house packed column using Reprosil-Pur C18-AQ (3 μm; Dr. Maisch GmbH, Germany). Buffer A was 0.1% formic acid and Buffer B was 0.1% formic acid in 80% acetonitrile. Peptides were resolved using a two-step linear gradient from 5 to 33% B over 45 min, then from 33 to 90% B over 10 min at a flow rate of 300 nL/min. The HPLC was coupled online to an Orbitrap Elite mass spectrometer operating in the positive mode using a Nanospray Flex™ Ion Source (Thermo Scientific) at 2.3 kV. Two full MS scans (m/z 300–1100) were acquired in the orbitrap mass analyzer with a resolution of 120,000 (at 200 m/z) every 8 DIA MS/MS events using isolation windows of 50 m/z each (e.g., 300–350, 350–400, …,650–700). MS/MS spectra were acquired in the ion trap operating in normal mode. Fragmentation was performed using collision-induced dissociation (CID) in the ion trap mass analyzer with a normalized collision energy of 35. AGC target and maximum injection time were 10e6 and 50 ms for the full MS scan, and 10e4 and 150 ms for the MS/MS can, respectively. Raw files were analyzed using EpiProfile.

**Cell proliferation assays.** Cell lines were plated at low confluency in either 12- or 24-well plates. Cell number was counted every 7 days using a Cellometer Auto T4 bright field cell counter (Nexcelom Bioscience). Cell index is reported as an average cell count of at least three biological replicates, normalized to a control group. For CRISPR-edited lines, a minimum of two distinct clones per group were included in experiments.

**Drug sensitivity assay.** Cells were plated at a density of 7000 cells per well in 96-well plate. They were treated with an increasing dose of EZH2 inhibitors, GSK343 and UNC1999, obtained from the Structural Genomic Consortium, ranging from 0.5 to 10uM. DMSO was used as a vehicle control. Cells were incubated with the drugs for 7 days and media was changed every 3 days. In order to assess cell viability, Alamar blue was added on cells on the seventh day for 6 h and absorbance at 570 nm and 600 nm was determined using i-Control microplate reader software by Tecan. Ratio of cell viability was calculated according to the following formula for measuring cytotoxicity and proliferation:

Percentage difference between treated and control cells

$$\frac{(O2 * A1) - (O1 * A2)}{(O2 * P1) - (O1 * P2)} * 100$$

where, O1 indicates molar extinction coefficient (E) of oxidized Alamar Blue (Blue) at 570 nm (80586), O2 indicates E of oxidized Alamar Blue at 600 nm

(117216), A1 absorbance of test wells at 570 nm, A2 absorbance of test wells at 600 nm, P1 absorbance of positive growth control well (cells plus Alamar Blue but no test agent) at 570 nm, P2 absorbance of positive growth control well (cells plus Alamar Blue but no test agent at 600 nm.

Graphs were plotted using Graphpad software using mean of three different replicates. Bars represent standard error of means.

**Mouse orthotopic xenograft.** All mice were housed, bred, and subjected to listed procedures according to the McGill University Health Center Animal Care Committee and were in compliance with the guidelines of the Canadian Council on Animal Care. Female NOD SCID mice (4–6 weeks) were used for xenograft experiments. Mice were injected with the following cell lines at a density of $10^5$ cells in the caudate putamen for BT245: (1) BT245 parental cells ($n = 9$), (2) BT245 Clone C1 (unedited cells, $n = 9$), (3) BT245 Clone C4 (K27M-KO clone, $n = 9$), (4) BT245 Clone D2 (K27M-KO clone, $n = 10$), and $7.10^5$ cells in the pons for DIPGXIII: (1) DIPGXIII parental cells ($n = 3$), (2) DIPGXIII K27M-KO ($n = 3$). The Robot Stereotaxic machine from Neurostar was used for stereotaxic injections. Mice were monitored daily for over a year, for neurological symptoms of brain tumors: weight loss, epilepsy, altered gait, lethargy. Mice brains were imaged by MRI when symptoms appeared. They were euthanized immediately when clinical endpoint is reached. The brains were removed and put in formalin for histology. Kaplan–Meier curve for mice survival was generated using the Graphpad software. Mice that died due to a tumor are considered as 1. Those that were still surviving at the end of the experiment, or those that were euthanized for non-tumor related reasons were considered as 0.

**Droplet digital PCR.** RNA was extracted from cells using the Aurum Total RNA Mini Kit (Bio-Rad) and concentration was quantified on the BioDrop uLite (Montreal Biotech). cDNA was generated using iScript Reverse Transcription Supermix (Bio-Rad). Target concentration was determined using the QX200 ddPCR EvaGreen Supermix assay (Bio-Rad) using 20 uL per reaction containing 0.5 ng of cDNA, using manufacturer's protocol cycling conditions with a 58 degrees annealing temperature and 40 cycles. Droplets were assayed using the QX200 Droplet Reader (Bio-Rad) and manually scored for positive signal using QuantaSoft Software (Bio-Rad). The concentration of positive droplets per target was normalized to the concentration of GAPDH. The relative mRNA abundance is shown as the average of three biological replicates (distinct passages of each cell line) determined by a single technical replicate. Primer sequences for each target are found in Supplementary Table 7.

**ChIP-sequencing.** Cells (cell lines or dissociated tumor cells) were fixed with 1% formaldehyde (Sigma). Fixed cell preparations were washed, pelleted and stored at −80 °C. Sonication of lysed nuclei (lysed in a buffer containing 1% SDS) was performed on a BioRuptor UCD-300 for 60 cycles, 10 s on 20 s off, centrifuged every 15 cycles, chilled by 4 °C water cooler. Samples were checked for sonication efficiency using the criteria of 150–500 bp by gel electrophoresis. After the sonication, the chromatin was diluted to reduce SDS level to 0.1% and before ChIP reaction 2% of sonicated drosophila S2 cell chromatin was spiked-in the samples for quantification of total levels of histone mark after the sequencing (see below).

ChIP reaction for histone modifications was performed on a Diagenode SX-8G IP-Star Compact using Diagenode automated Ideal ChIP-seq Kit. Twenty-five microliter Protein A beads (Invitrogen) were washed and then incubated with antibodies (anti-H3K27M (1:66, Millipore ABE419), anti-H3K27me3 (1:40, CST 9733), anti-H3K27me3 (1:100, Active Motif 61017), anti-H3K27me2 (1:50, CST 9728), anti-H3.3 (1:66, Millipore 09–838), anti-HA(1:100, CST 3724)) as listed in Supplementary Table 6, and 2 million cells of sonicated cell lysate combined with protease inhibitors for 10 h, followed by 20 min wash cycle with provided wash buffers.

ChIP reaction for SUZ12 and RING1B was performed as follows: antibodies (anti-SUZ12 (1:150, CST 3737), anti-RING1B (1:200, Active Motif 39663)), also listed in Supplementary Table 6) were conjugated by incubating with 40 ul protein A or G beads at 4 °C for 6 h, then chromatin from ~4 million cells was added in RIPA buffer, incubated at 4 °C o/n, washed using buffers from Ideal ChIP-seq Kit (one wash with each buffer, corresponding to RIPA, RIPA + 500 mM NaCl, LiCl, TE), eluted from beads by incubating with Elution buffer for 30 min at room temperature.

Reverse cross linking took place on a heat block at 65 °C for 4 h. ChIP samples were then treated with 2 ul RNase Cocktail at 65 °C for 30 min followed by 2 ul Proteinase K at 65 °C for 30 min. Samples were then purified with QIAGEN MiniElute PCR purification kit as per manufacturers' protocol. In parallel, input samples (chromatin from about 50,000 cells) were reverse crosslinked and DNA was isolated following the same protocol.

Library preparation was carried out using Kapa HTP Illumina library preparation reagents. Briefly, 25 ul of ChIP sample was incubated with 45 ul end repair mix at 20 °C for 30 min followed by Ampure XP bead purification. A tailing: bead bound sample was incubated with 50 ul buffer enzyme mix for 30 °C 30 min, followed by PEG/NaCl purification. Adaptor ligation: bead bound sample was incubated with 45 ul buffer enzyme mix and 5 ul of different TruSeq DNA adapters (Illumina) for each sample, for 20 °C 15 min, followed by PEG/NaCl purification

(twice). Library enrichment: 12 cycles of PCR amplification. Size selection was performed after PCR using a 0.6 × /0.8x ratio of Ampure XP beads (double size selection) set to collect 250–450 bp fragments.

ChIP libraries were sequenced using Illumina HiSeq 2000, 2500, or 4000 at 50 bp single reads.

**RNA-seq.** Total RNA was extracted from cell pellets and mouse tumors using the AllPrep DNA/RNA/miRNA Universal Kit (Qiagen) according to instructions from the manufacturer. Library preparation was performed with ribosomal RNA (rRNA) depletion according to instructions from the manufacturer (Epicentre) to achieve greater coverage of mRNA and other long non-coding transcripts.

Paired-end sequencing (100 bp) was performed on the Illumina HiSeq 2500 or 4000 platform.

**Whole-genome bisulphite sequencing.** Whole-genome sequencing libraries were generated from 700 to 1000 ng of genomic DNA spiked with 0.1% (w/w) unmethylated λ DNA (Promega) previously fragmented to 300–400 bp peak sizes using the Covaris focused-ultrasonicator E210. Fragment size was controlled on a Bioanalyzer DNA 1000 Chip (Agilent) and the KAPA High Throughput Library Preparation Kit (KAPA Biosystems) was applied. End repair of the generated dsDNA with 3′ or 5′ overhangs, adenylation of 3′ ends, adaptor ligation, and clean-up steps were carried out as per KAPA Biosystems' recommendations. The cleaned-up ligation product was then analyzed on a Bioanalyzer High Sensitivity DNA Chip (Agilent) and quantified by PicoGreen (Life Technologies). Samples were then bisulfite converted using the Epitect Fast DNA Bisulfite Kit (Qiagen) according to the manufacturer's protocol. Bisulfite-converted DNA was quantified using OliGreen (Life Technologies) and, based on quantity, amplified by 9–12 cycles of PCR using the Kapa Hifi Uracil + DNA polymerase (KAPA Biosystems) according to the manufacturer's protocol. The amplified libraries were purified using Ampure XP Beads (Beckman Coulter), validated on Bioanalyzer High Sensitivity DNA Chips, and quantified by PicoGreen.

Sequencing of the WGBS libraries was performed on the Illumina HiSeq2500/HiSeqX system using 125 or 150-bp paired-end sequencing.

**Analysis of ChIP-seq data.** Raw reads were aligned to human (UCSC hg19) or mouse (UCSC mm10) and Drosophila (UCSC dm6) genome build using BWA[45] version 0.7.17 with default parameters.

We divided genome into three different types of bins: 1 kb, 10 kb, and 100 kb, and counted number of reads from ChIP-seq experiments in those bins. We also counted reads in CpG islands (CGIs), promoters, and genic regions. All the read counting was done using bedtools[46] version 2.22.1. The annotations of CGIs, promoters, and RefSeq transcripts for hg19 and mm10 genomes were downloaded from UCSC Table Browser. Promoters were defined as 5 kb regions centered on RefSeq TSS. For genic regions, we took the region with the longest length of (TES-TSS) if multiple TSS and/or TES exist.

Top 1% bin scores were calculated as follows. After alignment, genome was divided into 1 kb bins. Then ChIP-seq RPKM values were calculated for each bin (for H3K27me3 and input), and input was subtracted (i.e., H3K27me3−input). The bins were sorted by these values, and average value of top 1% bins was taken. The data manipulation was performed in R.

As known, ChIP-seq enrichments are qualitative measures and cannot be directly used to compare signal strength across samples. This is particularly problematic where the overall level of the bound protein varies across conditions, as is the case for most of the histone modifications in this study. To address this issue, we quantified the ChIP enrichments for each sample using a technique called ChIP with reference exogenous genome (ChIP-Rx), which applies spike-in Drosophila chromatin as internal control[47]. For each ChIP-seq profile, we calculated the ChIP-Rx ratio (denoted as Rx) as follows:

$$Rx = \frac{s/s\_dmel}{i/i\_dmel}$$

where $s$ is the percentage of reads mapped to human or mouse genome in the target sample, $s\_dmel$ is the percentage of spike-in Drosophila genome in the sample, and similarly $i$ and $i\_dmel$ are defined for the input sample. To compare the genome-wide H3K27me3 distribution between H3K27M and H3WT cells, we normalized the tracks using ChIP-Rx ratios as follows: for each sample, the read counts in the genomic compartments (e.g., bins) are first divide by the total number of reads, then multiplied by ChIP-Rx ratio of the sample, and then multiplied by a normalization factor (which is the same for all samples, set as $10^{10}$ here) to avoid very small values.

We also normalized the ChIP-seq enrichments of target samples by their input data. Let $Si$ and $Ni$ be the read counts in the $i$-th genomic compartment of the sample and input, respectively, and TS and TN be the total read counts of the two samples, the normalization on $Si$ was done as follows:

$$Si\_norm = \log_2 \frac{Si/TS}{Ni/TN}$$

To avoid 0 in denominator or logarithm, we added a pseudo count (denoted as $c$,

here we set $c = 1$) during input-normalization, as follows:

$$Si\_norm = \log_2 \frac{(Si + c)/TS}{(Ni + c)/TN}$$

ChIP-sequencing coverage tracks were visualized using IGV 2.3 software[48,49]. To call narrow peaks, we used "macs2 callpeak -t ${CHIP} -c ${INPUT} -f BAM -g hs -n ${NAME} -q 0.01–outdir ${OUTDIR}". For broad histone marks, we call peaks using "macs2 callpeak -t ${CHIP} -c ${INPUT} -f BAM–broad -g hs–broad-cutoff 0.1 -n ${NAME}–outdir ${OUTDIR}".

Peak-calling was done using MACS2[50] version 2.1.1.

The SUZ12 peak-centered H3K27me2, H3K27me3 enrichment plots, and PMD plot were generated using ngs.plot.r package[51]. The signal around SUZ12 peak centers ($+/-50$ kb) was normalized by input and by ChIP-Rx ratio. The signal around PMD regions was extended by 100 kb on each side. The following command was used to generate the plots: "./ngs.plot.r -G hg19 -R bed -C {Config file} -O {output file} -L {extend by number of bases}". The config file contains the names of H3K27me3/H3K27me2 and input BAM files and BED file name of SUZ12 peaks for SUZ12-cenetered plot. For PMD plot, the config file contains the BED file of PMD regions instead.

The aggregate plots of SUZ12, H3K27me2, and H3K27me3 for primary cell lines were generated using deepTools v3.1.0[52]. The signals were normalized by both input and ChIP-Rx ratios as follows:

$$Si\_norm = \log_2 \frac{(Rx * (Si + c))/TS}{(Ni + c)/TN}$$

where Rx is the ChIP-Rx ratio, the rest variables have the same meanings as in the input-normalization equation. The normalization was done using deepTools' bamCompare, where the bin size and fragment length were set to 500 bp and 200 bp, respectively. After normalization, the signals were aggregated around ($+/-50$ kb) the centres of the top CGIs (for SUZ12) or TSS of repressed genes (for H3K27me3) using deepTools' computeMatrix and visualized by averaging the score in bins equidistant from the mid-point (with NaN values discarded).

Heatmap plots for comparing H3K27me3, SUZ12, RING1B, and DNA methylation were generated using ChAsE v.1.0.11 software[53]. The regions were centered on CGIs, extending 5 kb, 10 kb, or 50 kb upstream and downstream.

We characterized the variations of H3K27me3 at CGIs in H3WT (or H3K27M-KO) versus H3K27M cells by plotting the differential enrichment of H3K27me3 between the two conditions against the H3K27me3 enrichment in H3WT (or H3K27M-KO) cells. Accordingly, the $x$ and $y$ axes are defined as follows:

$$X = \log_2\left(\frac{H3WT}{INPUT}\right), \quad Y = \log_2\left(\frac{H3K27M}{H3WT}\right)$$

where H3WT, H3K27M, and INPUT represent the processed read counts at CGIs in the three types of samples, i.e., WT, K27M, and input ChIP-seq libraries. The "processed read counts" here means the number of mapped reads at CGIs divided by total read counts of the genome. Note that for H3WT and H3K27M, we also multiplied the processed read counts by their ChIP-Rx ratios. We filtered out the CGIs, which overlap with the ENCODE DAC blacklisted regions[54]. To reduce stochasticity in low coverage regions, we took the average of H3K27me3 enrichment in two available WT samples. Therefore, the $x$ and $y$ axes are modified to:

$$X = \frac{1}{2}\left(\log_2\left(\frac{H3WT\_1}{INPUT\_1}\right) + \log_2\left(\frac{H3WT\_2}{INPUT\_2}\right)\right),$$
$$Y = log2(H3K27M) - 1/2 * (log2(H3WT_1) + log2(H3WT_2))$$

The plot of H3K27me3 variations at promoters is similar to the one at CGIs. The dots were colored based on the differential analysis of gene expression (see Analysis of RNA-seq data for details). The H3K27me3 variations at promoters are categorized into four groups: Gained in K27M, Absent, Lost in K27M, and Retained. The categorization was based on H3K27me3 values in WT ($x$-axis) and H3K27me3 difference ($y$-axis) as follows: for CGI/DNA methylation plot (Fig. 2g, Supplementary Fig. 10g–h, 13a–c): Gained in K27M (BT245: $y > 0$; DIPGXIII: $y > 2.6$), Absent (BT245: $y < 0$, $x < -3.75$; DIPGXIII: $y < 2.5$, $x < -5$), Lost in K27M (BT245: $y < -4$, $x > -2.5$; DIPGXIII: $y < -2.5$, $x > -2.5$), Retained (BT245: $-3 < y < 0$, $x > 1.25$; DIPGXIII: $y > -2$, $x > 0.5$); for promoter/differential expression plot (Fig. 4a): Gained in K27M (BT245: $y > 0$), Absent (BT245: $y < 0$, $x < -2.5$), Lost in K27M (BT245: $y < -4$, $x > -2.5$), Retained (BT245: $-3 < y < 0$, $x > 0$).

To show the correlation between H3K27me2 in K27M and H3K27me3 in wild-type isogenic cells, we first sorted the 1 kb non-overlapping bins based on the H3K27me2 enrichment (after input normalization) in K27M cells, then grouped the sorted bins into 1000 windows, and plot the mean values of ChIP enrichment (WT-H3K27me3 and WT-H3K27me2) in the 1000 windows. We used the dashed line to indicate the K27M-H3K27me2 enrichment per window. The procedure was adapted from ref. [55].

**Analysis of RNA-seq data.** Raw reads were aligned to human genome build (UCSC hg19) using STAR[56] version 2.5.3a. For each sample, we counted the mapped reads for each gene from annotation files in GTF format (downloaded from UCSC Table Browser) using featureCounts program (version 1.5.3).

Read counts from RNA-seq alignment were transformed (regularized log transformation) using the rlog function from R package DESeq2. Differential expressed (DE) genes between H3WT and H3K27M were defined as genes with log2FoldChange larger (Up genes) or smaller (Down genes) than 1 and with adjusted $p$-value smaller than 0.05.

To make Decile plot of the DE genes, first we categorized the DE genes into ten groups based on their expression levels in parental samples (i.e., before CRISPR editing or overexpression). Then we counted the number of genes (including Up genes and Down genes) in each category, and plotted the gene counts using geom_bar in R Package ggplot2.

Gene set enrichment analysis (GSEA) was performed using the GSEA tool from Broad Institute (https://software.broadinstitute.org/gsea/index.jsp)[57,58].

**Analysis of WGBS data.** Raw reads were aligned to human genome build (UCSC hg19) using BWA (version 0.6.1)[59] after converting the reference genome to bisulfite mode. Low-quality sequence at the 3′ ends were trimmed. And for the overlapping paired-end reads, we clipped the 3′ end of one of them to avoid double counting. This was done on both forward and reverse strand. After alignment, we filtered reads that were duplicate, or were poorly mapped (with more than 2% mismatches), or not mapped at the expected distance based on the library insert size.

To call methylation of individual CpGs, Samtools[59] (version 0.1.18) in mpileup mode was applied. CpGs covered by less than five reads were removed. We also discarded the CpGs that were overlapped with SNPs from dbSNPs (137) or were located within the ENCODE DAC blacklisted regions or Duke excluded regions[54].

To call partially methylated domains (PMDs), we used a window size of 10 kb and slid the window on the measurements of methylation level at each CpG. For each 10 kb window where the average methylation level of mCGs was less than 70% (with each CpG covered by at least 5 MethylC-Seq reads), we extended the region with increments of 10 kb until the average methylation level of the region was greater than 70%. After that, we reported the regions with size larger than 1 Mb as PMDs.

**Reporting summary.** Further information on experimental design is available in the Nature Research Reporting Summary linked to this article.

## Code availability

Custom R scripts used to generate the top 1% bin scores, the plots of H3K27me3 variations, the correlation plots of K27M-K27me2 and WT-K27me3 and the Decile plots are available upon request.

## Data availability

The sequencing data reported in this paper are available for download through the GenAP portal at https://datahub-jv6f4mbl.udes.genap.ca/. All other relevant data supporting the key findings of this study are available within the article and its Supplementary Information files or from the corresponding author upon reasonable request. The source data underlying Figs. 1a, 1d, 2a, 2d, 3d-e, 5a and Supplementary Figs 2, 4a, 7a, 8b-c, 8e-f, 9a-d, 10b-e, 11b-e, 13d, 16c-d, 17a-e, 18a-b, 19b and 20a-c are provided as a Source Data file. A reporting summary for this Article is available as a Supplementary Information file.

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

## Acknowledgements

This work was supported by funding from: US National Institutes of Health (NIH grant P01-CA196539 to N.J., J.M., P.W.L, B.A.G., T.P.; GM110174 to B.A.G; T32GM008275 and TL1TR001880 to D.M.M.), the Canadian Institutes for Health Research (CIHR grant MOP-286756 and FDN-154307 to N.J., EP1–120608 to T.P. and P.J.T.-156086 to C.L.K.), the Fonds de Recherche du Québec en Santé (FRQS) salary award to C.L.K. N.J. is a member of the Penny Cole Laboratory and the recipient of a Chercheur Boursier, Chaire de Recherche Award from the FRQS. This work was performed within the context of the International CHildhood Astrocytoma INtegrated Genomic and Epigenomic (ICHANGE) consortium, and the Stand Up to Cancer, Canada Cancer Stem Cell Dream Team initiative, with funding from Genome Canada and Genome Quebec. A.S.H., W.A.C., and N.D.J. are recipients of fellowships from FRQS. D.B. is the recipient of a fellowship from the TD CanadaTrust/ Montreal Children's Hospital Foundation. M.K.M. is funded by a CIHR Banting post-doctoral fellowship. P.W.L. is a Pew Scholar in the Biomedical Sciences. C.L. acknowledges support from Damon Runyon Cancer Research Foundation and Matthew Larson Foundation. P.S. and M.P. are supported by the ERC (H3.3Cancer). We thank Alexey Soshnev for providing the comprehensive schema for the K27M mechanism of action. We are especially grateful for the generous philanthropic donations of Kat D DIPG and We Love You Connie Foundations.

## Author contributions

A.S.H. and B.K. led and performed a majority of the functional studies, and were actively involved in study design, data analysis, interpretation, and manuscript preparation. H.C., S.P.-C., N.D.J., R.L., H.N., B.H., G.C. and W.A.C. contributed to bioinformatic analysis of the data and interpretation of the results. M.Z., S.D., C.C.L.C., J.B., Ab.M., D.B., D.F., M.K.M. and S.U.J. contributed to data collection, analysis, and study design. D.M.M. and B.G. led the histone proteomics experiments and analysis. L.M. contributed to study design, data interpretation, and manuscript preparation. B.E. and A.W. assisted with the collection of patient samples, study design, and data interpretation. M.P. and P.S. generated the mouse model

used for some ChIP-seq experiments, Al.M., T.P., C.L., P.W.L., B.G., C.L.K, N.J. and J.M. contributed to study design, data interpretation, and manuscript preparation.

## Additional information

**Competing interests:** The authors declare no competing interests

