## [Peer Review File · Nature Communications]

REVIEWERS' COMMENTS:

Reviewer #1 (Remarks to the Author):

We have carefully read the author's point-by-point answer to the reviewers' comments. Overall, this addresses our comments and we therefore recommend publication of this manuscript.

Reviewer #2 (Remarks to the Author):

In this study, the authors use a combination of mass spec and reference normalized ChIP-seq to show that H3K27me_{2/3} levels are reduced and have an altered distribution pattern in pediatric glioma cells carrying an H3K27M mutation, as well as in cells that have been edited to contain the mutation. They also show that inhibition of EZH1/EZH2 with UNC1999 does not phenocopy the effect of H3K27M, although H3K27M mutants have only a modest effect on gene expression, similar to loss of PRC2 activity. Finally, they show that the presence of H3K27M is essential for tumorigenesis in an orthotopic xenograft model.

Overall, the experiments are very well performed and mostly support the authors conclusions. However, I am not completely convinced that the authors have provided as clear a mechanism for tumorigenesis as they seem to believe. That said, they don't overstate their conclusions and their data is much more clear than other studies published on this topic. In addition, whatever the exact mechanism, it seems clear that the presence of the H3K27M mutation is responsible for inducing a differentiation block. Although it is not clear exactly how H3K27me₃ (and to a lesser extent me₂) redistribution causes this differentiation block, this is still a very interesting observation and it suggests that "spreading" of H3K27me₃ into intergenic sites is an important aspect of regulating gene expression in gliomas. It also seems likely (although not proven, see below) that this gene expression dysregulation is what causes the differentiation block.

Minor points

1. The SUZ12 peak centered aggregate plots in various genome edited or other cell lines (Supp Fig 7c, 8g and 10f) give apparently different profiles than that seen in the primary tissue samples (Supp Figure 3d). Is there an explanation for this? This should be noted in the text.

2. What is still not entirely clear (although it seems very plausible), is whether dysregulated gene expression is a key aspect of H3K27M function, or if perhaps H3K27me_{2/3} distribution changes have some other structural role (for example, studies in *Drosophila* suggest that loss of PRC2 activity can impact proper chromosome segregation during cell division). That said, I think completely nailing down the mechanism is beyond the scope of the current study. However, in the absence of including data that shows the existence of the differentiation block (my understanding is that this data is included in a second submitted paper), it would be useful to know if any of the dysregulated genes contribute at all to tumor cell growth. It is interesting that there is low level dysregulation of genes important for stemness and neurogenesis. It is unlikely that a single target is responsible for the differentiation block. I am hesitant to suggest more experiments in what is already a very extensive body of work. However, it would be useful to know if knockdowns of some key gene targets (e.g. ID genes) in H3K27M mutants at all impairs cell growth.

NCOMMS-18-34515A

REVIEWERS' COMMENTS:

Reviewer #1 (Remarks to the Author):

We have carefully read the author's point-by-point answer to the reviewers' comments. Overall, this addresses our comments and we therefore recommend publication of this manuscript.

We would like to thank the reviewer for positive appraisal of our revisions and manuscript overall.

Reviewer #2 (Remarks to the Author):

In this study, the authors use a combination of mass spec and reference normalized ChIP-seq to show that H3K27me2/3 levels are reduced and have an altered distribution pattern in pediatric glioma cells carrying an H3K27M mutation, as well as in cells that have been edited to contain the mutation. They also show that inhibition of EZH1/EZH2 with UNC1999 does not phenocopy the effect of H3K27M, although H3K27M mutants have only a modest effect on gene expression, similar to loss of PRC2 activity. Finally, they show that the presence of H3K27M is essential for tumorigenesis in an orthotopic xenograft model.

Overall, the experiments are very well performed and mostly support the authors conclusions. However, I am not completely convinced that the authors have provided as clear a mechanism for tumorigenesis as they seem to believe. That said, they don't overstate their conclusions and their data is much more clear than other studies published on this topic. In addition, whatever the exact mechanism, it seems clear that the presence of the H3K27M mutation is responsible for inducing a differentiation block. Although it is not clear exactly how H3K27me3 (and to a lesser extent me2) redistribution causes this differentiation block, this is still a very interesting observation and it suggests that "spreading" of H3K27me3 into intergenic sites is an important aspect of regulating gene expression in gliomas. It also seems likely (although not proven, see below) that this gene expression dysregulation is what causes the differentiation block.

We would like to thank the reviewer for thorough assessment and appraisal our revised manuscript.

Minor points

1. The SUZ12 peak centered aggregate plots in various genome edited or other cell lines (Supp Fig 7c, 8g and 10f) give apparently different profiles than that seen in the primary tissue samples (Supp Figure 3d). Is there an explanation for this? This should be noted in the text.

The data of tumor tissue ChIP-seq is not normalized by drosophila spike-in (no spike-in was used in the ChIPs), therefore K27M sample profiles are higher than wild-type tumors. In addition, please note that we used SUZ12 peak data from the overlap of K27M and WT cell line SUZ12 peaks, which is not matching to tumor tissue samples. This information is added to the figure legend. We also revised the plot to include 50 kb flanking regions on each side of the peak centre (in the original figure it was 10 kb), in order to match our other SUZ12 peak centered aggregate plots. We would like to note that the tumor and cell line data should not be expected to look identical, however, they both illustrate the same trend: greatly reduced overall K27me3 levels, and much higher proportion of K27me3 concentrated around PRC2 nucleation sites in the presence of the H3K27M mutation.

2. What is still not entirely clear (although it seems very plausible), is whether dysregulated gene expression is a key aspect of H3K27M function, or if perhaps H3K27me2/3 distribution changes have some other structural role (for example, studies in *Drosophila* suggest that loss of PRC2 activity can impact proper chromosome segregation during cell division). That said, I think completely nailing down the mechanism is beyond the scope of the current study. However, in the absence of including data that shows the existence of the differentiation block (my understanding is that this data is included in a second submitted paper), it would be useful to know if any of the dysregulated genes contribute at all to tumor cell growth. It is interesting that there is low level dysregulation of genes important for stemness and neurogenesis. It is unlikely that a single target is responsible for the differentiation block. I am hesitant to suggest more experiments in what is already a very extensive body of work. However, it would be useful to know if knockdowns of some key gene targets (e.g. ID genes) in H3K27M mutants at all impairs cell growth.

We would like to thank reviewer for valuable ideas on the follow-up of our findings. We agree with the reviewer that these additional experiments are beyond the scope of current study, especially since there is already extensive amount of data included. We definitely plan to follow up in these directions in future.